# Fatal Dengue, Chikungunya and Leptospirosis: The Importance of Assessing Co-infections in Febrile Patients in Tropical Areas

**DOI:** 10.3390/tropicalmed3040123

**Published:** 2018-11-26

**Authors:** Jaime A. Cardona-Ospina, Carlos E. Jiménez-Canizales, Heriberto Vásquez-Serna, Jesús Alberto Garzón-Ramírez, José Fair Alarcón-Robayo, Juan Alexander Cerón-Pineda, Alfonso J. Rodríguez-Morales

**Affiliations:** 1Public Health and Infection Research Group, Faculty of Health Sciences, Universidad Tecnológica de Pereira, Pereira 66003, Colombia; jaancardona@utp.edu.co (J.A.C.-O.); caedjimenez@utp.edu.co (C.E.J.-C.); 2Infection and Immunity Research Group, Faculty of Health Sciences, Universidad Tecnológica de Pereira, Pereira 66003, Colombia; 3Secretary of Health of Ibagué, Ibague, Tolima 730001, Colombia; herivasquez@gmail.com; 4School of Medicine, Faculty of Health Sciences, Universidad de Tolima, Ibagué, Tolima 730001, Colombia; jaancardona@gmail.com (J.A.G.-R.); arodriguez@jidc.org (J.A.C.-P.); 5Secretary of Health of Tolima, Ibague, Tolima 730001, Colombia; okdjx@hotmail.com; 6Medical School, Universidad Privada Franz Tamayo UNIFRAZ, Cochabamba 4780, Bolivia

**Keywords:** dengue, chikungunya, *Leptospira*, co-infection, Colombia, Latin America

## Abstract

The febrile patient from tropical areas, in which emerging arboviruses are endemic, represents a diagnostic challenge, and potential co-infections with other pathogens (i.e., bacteria or parasites) are usually overlooked. We present a case of an elderly woman diagnosed with dengue, chikungunya and *Leptospira interrogans* co-infection. Study Design: Case report. An 87-year old woman from Colombia complained of upper abdominal pain, arthralgia, myalgia, hyporexia, malaise and intermittent fever accompanied with progressive jaundice. She had a medical history of chronic heart failure (Stage C, New York Heart Association, NYHA III), without documented cardiac murmurs, right bundle branch block, non-valvular atrial fibrillation, hypertension, and chronic venous disease. Her cardiac and pulmonary status quickly deteriorated after 24 h of her admission without electrocardiographic changes and she required ventilatory and vasopressor support. In the next hours the patient evolved to pulseless electrical activity and then she died. Dengue immunoglobulin M (IgM), non-structural protein 1 (NS1) enzyme-linked immunosorbent assay (ELISA), microagglutination test (MAT) for *Leptospira interrogans* and reverse transcription polymerase chain reaction (RT-PCR) for chikungunya, were positive. This case illustrates a multiple co-infection in a febrile patient from a tropical area of Latin America that evolved to death.

## 1. Introduction

The arrival of emerging arboviruses, such as chikungunya and Zika, in Latin America has extended the diagnostic spectrum of acute febrile patients. Although there is no specific management for Zika or chikungunya infections, the importance of etiologic diagnosis relies on the need for proper follow-up of those patients that evolve to chikungunya chronic disease [1,2], and in case of sexual transmission of transmission through body fluids in the case of Zika, and possibly other arboviruses [3]. The clinical picture is difficult to differentiate between arbovirus infections, and co-infections could be more frequent than expected [4,5]. Moreover, co-infections with other infectious microorganisms can occur, but they are less frequently reported. These infectious agents may require different and specific management strategies and assessment [6], and their impact on immune pathogenesis and clinical outcome is still unknown. However, simultaneous infections with arbovirus and other microorganisms like *Leptospira* spp. have evolved to death [6,7,8]. Hence, arbovirus co-circulation with other endemic bacteria and protozoa complicates the necessary laboratory etiological assessment of febrile patients in tropical settings.

## 2. Case Presentation

An 87-year old woman from Junín, a rural area of the municipality of Venadillo, Tolima, Colombia, endemic for dengue, chikungunya and Zika [9,10], consulted at the local hospital. She complained of upper abdominal pain, arthralgia, myalgia, hyporexia, malaise and intermittent fever accompanied with progressive jaundice. She had a medical history of chronic heart failure (Stage C, New York Heart Association, NYHA III), without documented cardiac murmurs, right bundle branch block, non-valvular atrial fibrillation, hypertension, and chronic venous disease. On physical examination, she was conscious and had tachycardia, tachypnea, mucocutaneous jaundice, venous neck pulsations with abdominojugular reflux, increased S1 intensity with irregular rhythm, a systolic murmur at both upper sternal borders, and diminished breath sounds at both lung bases. Abdominal and neurological findings were unremarkable, with a non-painful palpable liver 3 cm down the costal border at the mid-clavicular line. Haematological evaluation showed leucopenia, and thrombocytopenia; other test results are shown in Table 1.

The electrocardiographic evaluation showed an atrial fibrillation with rapid ventricular response and right bundle branch block. The chest X ray showed cardiomegaly and bilateral pleural effusion. The patient was initially managed as severe dengue with hepatic compromise and she was transferred to an intensive care unit (ICU). After her admission in ICU she had one episode of hypoglycemia and her renal function gradually worsened. Alongside supportive treatment, antibiotic therapy with cephazolin was initiated. Her cardiac and pulmonary status quickly deteriorated after 24 h of admission without electrocardiographic changes, and she required ventilatory and vasopressor support. In the next hours, the patient evolved to pulseless electrical activity and died.

Blood samples were tested at the Public Health Laboratory of Tolima. Dengue IgM-antibodies (44.4% sensitivity, 99.1 specificity) and non-structural protein 1 (NS1) dengue protein through enzyme-linked immunosorbent assay (ELISA) (93.9% sensitivity, 97.4% specificity) were both positive [11]. Additionally, titers against *Leptospira interrogans* (serogroup Tarassovi, serovar Tarassovi, 1:400) were detected through non-paired microagglutination test (MAT) (93.8% sensitivity, 90.4% specificity) [12]. Chikungunya infection was confirmed with reverse transcription polymerase chain reaction (RT-PCR) (90% sensitivity, 100% specificity) [13]. Zika RT-PCR was negative (Table 2). Necropsy was not performed in the patient.

## 3. Discussion

This case highlights the occurrence of co-infections in febrile patients from tropical areas of Latin America and Colombia. Outbreaks of leptospirosis and malaria concurrent to dengue outbreaks have been reported [14], and although co-infections were rare (<1%), previous reports have showed the evolution to fatal disease of coinfections of *Leptospira* spp. with arboviruses [6,7,8,15], and an association between co-infection and evolution to shock [16]. Notwithstanding, triple co-infection is rarely reported.

The diagnostic assessment of febrile patients in tropical areas remains a challenge, and there is a need for improvement of multi-diagnosis assessment. Although arboviral disease still lacks specific management beyond symptomatic relief, and fluid resuscitation and complication management in severe cases, co-infections with bacteria or protozoa could require specific therapeutic measures like chemotherapy. In the case of leptospirosis, early diagnosis and treatment are related with shorter disease duration [17]. Moreover, some arboviral diseases may require long-term follow-up for assessment of chronic disease, like chikungunya [1,2], or may represent a risk for transmission through non-vectorial ways, like Zika [3]. Hence, etiologic diagnosis will improve not only management but will also provide information regarding the need of additional measures beyond the febrile disease. Certainly, early antibiotic treatment when suspecting leptospirosis with coinfections, should be considered, and based on the clinical condition, prompt admission to the ICU. In patients living or proceeding from endemic areas where all of these agents are cocirculating, such coinfections should be considered and ruled out, especially in an elderly patient.

Currently, molecular test tools have been developed for multiplex detection of dengue, chikungunya and *Leptospira* [18], and for dengue, malaria and *Leptospira* [19]. Ideally, these tools should be available in tropical clinical settings where they are required for accurate and prompt diagnosis. On the other hand, new tools must be designed ensuring cost-effectivity and pointing also to newly emergent arboviral infections such as Mayaro [20], and other less frequent reported causes of undifferentiated febrile illness (UFI), like rickettsiosis or brucellosis in order for rapid detection of emerging outbreaks and reduce the extent of the ‘gray zone’ of UFI.

Notwithstanding, our report has some limitations regarding diagnostic assessment of the patient. Although chikungunya and dengue infection were confirmed through molecular and immunological means, respectively, paired samples for *Leptospira* infection confirmation were not possible. However, the patient comes from an endemic area, had a clinical picture compatible with Weil’s syndrome, and had high MAT titers (≥400). Although the significance of titers in a single serum specimen is a matter of considerable debate, in various areas with the proper clinical picture it could be considered as proof of current or recent leptospirosis [21].

On the other hand, heart ultrasonographic evaluation could not be conducted in order to assess the etiology of heart failure exacerbation and of the new-onset heart murmur. Although common in dengue fever, myocarditis is considered an unusual complication in leptospirosis, and reports are increasingly highlighting the importance of cardiovascular involvement during and after chikungunya infection [22,23,24]. In this setting myocarditis seems like a plausible diagnosis. Besides, because the patient complained about fever and a she had a new-onset heart murmur, infective endocarditis was an important differential diagnosis, as well as other non-assessed endemic infectious that cause fever and cardiomyopathy, like Chagas’ disease [25]. However, to conclude a diagnosis of associated myocarditis, creatine phosphokinase (CPK) elevations or post-mortem examination are also required in addition to electrocardiographical (EKG) and echocardiographical findings.

Regarding treatment, antibiotic therapy of this patient is a matter of discussion too. Although cephazolin and cephalexin have been proved in animal models for *Leptospira* treatment, apparently with good response, currently, they are not the treatment of choice [26,27]. Doxycycline, which is among the preferred antibiotic therapy, has showed reduction of chikungunya virus replication and reduction in serum levels of interleukin-6 (IL-6), tumoral necrosis factor (TNF) and mortality in in-vitro and clinical studies [28]. Since antibiotic therapy in leptospirosis is open to discussion because the lack of conclusive evidence of its benefits, especially penicillin in severe disease [27]. In the setting of an arboviral co-infection, it remains unclear if the possible immunomodulation and antiviral effect attained with doxycycline could enhance treatment benefits.

The clinical challenge remains complex and it can become even more difficult, since the risk of introduction and emergence of other infectious disease in different territories continue to exist. As has been recently suggested [8], during outbreaks of arbovirus infection, including dengue, chikungunya and Zika (as is currently happening in various countries of the Americas), the attack rate of infections is usually high, especially when it occurs in a population with low levels or no pre-existing immunity. Even if arboviral infections are confirmed by reference molecular testing, leptospirosis should be considered if other symptoms or laboratory abnormalities are suggestive [8,14]. Leptospirosis should be a major differential diagnosis of dengue-like illness, always considering the possibility of co-infection with arboviruses in endemic areas, and in travelers returning from these regions, especially individuals from rural areas who have contact with domestic and wild animals.

## Figures and Tables

**Table 1 tropicalmed-03-00123-t001:** Patient laboratory data.

Variable	Laboratory Reference Range, Adults	On ED Admission	9 h after ICU Admission	12 h after ICU Admission
Hematocrit (%)	35–50	39.8	-	-
Hemoglobin (g/dL)	11.0–16.5	13.1	-	-
White blood cell count (per mm^3^)	5.0–10	3.7	-	-
*Differential count* (%)				
Granulocytes	43–76	71.2	-	-
Lymphocytes	17–48	25.2	-	-
Monocytes	4.0–10	3.6	-	-
Platelet count (per mm^3^)	150,000–450,0000	68,000	-	-
Erythrocyte count (per mm^3^)	3,800,000–5,800,000	4,840,000	-	-
Erythrocyte sedimentation rate (mm/h)	0–20	1	-	-
Total bilirubin (mg/dL)	0–1.1	10.5	-	-
Indirect bilirubin (mg/dL)	0–0.75	3.48	-	-
Direct bilirubin (mg/dL)	0–0.25	7.02	-	-
Urea nitrogen (mg/dL)	4.7–23	29	-	-
Creatinine (mg/dL)	0.9–1.3	1.11	-	-
*Blood venous gases*				
Inspired fraction of oxygen	-	-	0.21	0.81
pH	7.3–7.4	-	7.41	6.96
Partial pressure of carbon dioxide (mm/hg)	38–50	-	21.5	38
Serum lactate (mg/dL)	-	-	4.3	14
Partial pressure of oxygen (mm/hg)	35–50	-	74	57
Serum glucose (mg/dL)	70–99	145	47	43

-: Not available; ED: Emergency department; ICU: Intensive care unit.

**Table 2 tropicalmed-03-00123-t002:** Patient infection laboratory data.

Agent and Test	Result
Dengue IgM-antibodies	Positive
NS1 protein ELISA for dengue	Positive
Microagglutination test (MAT) for *Leptospira interrogans*	Positive
Titers for serogroup Tarassovi, serovar Tarassovi	1:400
Chikungunya RT-PCR	Positive
Zika RT-PCR	Negative

IgM: Immunoglobulin M; ELISA: Enzyme-linked immunosorbent assay; NS1: Non-structural protein 1; RT-PCR: Reverse transcription polymerase chain reaction.

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
