# Peer review of "Fatal Dengue, Chikungunya and Leptospirosis: The Importance of Assessing Co-infections in Febrile Patients in Tropical Areas"

_tropicalmed, 2018, doi:10.3390/tropicalmed3040123_

Round 1

Reviewer 1 Report

Overall the paper is a nice contribution to the small but growing literature of presentation and management of persons with co-infections. 

The authors discuss the limitations of the various infectious diagnoses .

The authors allude to , but only marginally discuss, how this patient's presentation would or will not affect management of future patients. For example, would there be more rapid institution of penicillin based therapy knowing a febrile patient came from an leptospiral endemic region? Would supportive management for multiple infectious illness differ from supportive management of a single suspected infection? And finally, would the authors search for multiple causalities in future cases - or only for those with specific epidemiologic risks (and if the latter - what risks would be considered as relevant)?

Specific comments: 

Line 69: the phrase "worsened to pre-renal lesion" is not clear; and can be deleted without loss of meaning. 

Lines 111-118: the discussion regarding this patient's progressive cardiac decline is speculative and should be significantly shortened. A diagnosis of myocarditis from Leptospirosis is not supported with a finding of only new RBBB, in the absence of CPK elevations or post-mortem examination.  Moreover, such a finding would not be surprising as a complication of severe exacerbation of right heart failure due to multiple infections. 

Lines 119-127: the discussion regarding antimicrobial management can be shortened significantly as more complex decision making was not possible due to the patient's rapid demise. 

References: for a short report, 31 references seems excessive; would suggest that the authors review these for clear relevance to a very directed and concise manuscript

Author Response

Responses to Reviewer 1

Overall the paper is a nice contribution to the small but growing literature of presentation and management of persons with co-infections.

The authors discuss the limitations of the various infectious diagnoses.

Response: Thanks for your comments.

The authors allude to, but only marginally discuss, how this patient's presentation would or will not affect management of future patients. For example, would there be more rapid institution of penicillin-based therapy knowing a febrile patient came from a leptospiral endemic region? Would supportive management for multiple infectious illness differ from supportive management of a single suspected infection? And finally, would the authors search for multiple causalities in future cases - or only for those with specific epidemiologic risks (and if the latter - what risks would be considered as relevant)?

Response: We have added in the Discussion (lines 96-99), the following text:

Certainly, early antibiotic treatment when suspecting leptospirosis with coinfections, should be considered, and based on the clinical condition, prompt admission to the ICU. In patients living or proceeding from endemic areas where all of these agents are cocirculating, such coinfections should be considered and ruled-out.

Specific comments:

Line 69: the phrase "worsened to pre-renal lesion" is not clear; and can be deleted without loss of meaning.

Response: Agree. Done.

Lines 111-118: the discussion regarding this patient's progressive cardiac decline is speculative and should be significantly shortened. A diagnosis of myocarditis from Leptospirosis is not supported with a finding of only new RBBB, in the absence of CPK elevations or post-mortem examination.  Moreover, such a finding would not be surprising as a complication of severe exacerbation of right heart failure due to multiple infections.

Response: We did not affirmed that the patient had a myocarditis. But this, should be considered in those patients. However, we added the following in that paragraph: Although that, to conclude a diagnosis of associated myocarditis, CPK elevations or post-mortem examination are also required in addition to EKG and echocardiographical findings.

Lines 119-127: the discussion regarding antimicrobial management can be shortened significantly as more complex decision making was not possible due to the patient's rapid demise.

Response: Although the patient rapidly died, earlier antimicrobial and immunomodulatory treatment would probably change the evolution in these cases. Then, we consider this is necessary to keep.

References: for a short report, 31 references seems excessive; would suggest that the authors review these for clear relevance to a very directed and concise manuscript.

Response: We agree. We reduced in 3 references the number.

Reviewer 2 Report

In this manuscript, authors provide an very improtant muiltiple-infection cases, which should be paied more attentions, especially in undeveloped region. From the whole, this report deserves for publication here, and several point comments listed:

This is a death case of a 87-year old patient, aging factor should be considered as other infectious factors for evaluation the severity of multi-infections;

Diagnostic assessment is a main limition for the final results of patients, like in here, the patients even accepted no specific therapy, and authors talked the Zika virus many times here, is there any try to comfirm Zika virus infection? It is very eager for quit test-kit of Flavivirus co-infections in tropical area;

Please summary infection laboratory data in another table?

Author Response

Responses to Reviewer 2

In this manuscript, authors provide a very important multiple-infection cases, which should be paid more attentions, especially in undeveloped region. From the whole, this report deserves for publication here, and several point comments listed:

Response: Thanks for your comments.

This is a death case of a 87-year old patient, aging factor should be considered as other infectious factors for evaluation the severity of multi-infections;

Response: We have included it now in the text (line 99).

Diagnostic assessment is a main limitation for the final results of patients, like in here, the patients even accepted no specific therapy, and authors talked the Zika virus many times here, is there any try to confirm Zika virus infection? It is very eager for quit test-kit of Flavivirus co-infections in tropical area;

Response: Diagnosis on case like this should be rely more on molecular tools. CHIKV was diagnosed by RT-PCR and Zika was ruled-out in the same way. This has been clarified in the text (Line 79).

Please summary infection laboratory data in another table?

Response: Included.